# Implication of the Autophagy-Related Protein Beclin1 in the Regulation of EcoHIV Replication and Inflammatory Responses

**DOI:** 10.3390/v15091923

**Published:** 2023-09-14

**Authors:** Myosotys Rodriguez, Florida Owens, Marissa Perry, Nicole Stone, Yemmy Soler, Rianna Almohtadi, Yuling Zhao, Elena V. Batrakova, Nazira El-Hage

**Affiliations:** 1Department of Immunology and Nanomedicine, Herbert Wertheim College of Medicine, Florida International University, Miami, FL 33199, USA; myrodrig@fiu.edu (M.R.); fowens@fiu.edu (F.O.); mperr046@fiu.edu (M.P.); nston006@med.fiu.edu (N.S.); yemmysoler@yahoo.com (Y.S.); ralmohta@fiu.edu (R.A.); 2Center for Nanotechnology in Drug Delivery, University of North Carolina at Chapel Hill, Chapel Hill, NC 27599, USA; yulingz@email.unc.edu (Y.Z.); batrakova@unc.edu (E.V.B.); 3Eshelman School of Pharmacy, University of North Carolina at Chapel Hill, Chapel Hill, NC 27599, USA

**Keywords:** autophagy, Beclin1, EcoHIV, intranasal delivery, mannosylated polyethyleneimine

## Abstract

The protein Beclin1 (BECN1, a mammalian homologue of ATG6 in yeast) plays an important role in the initiation and the normal process of autophagy in cells. Moreover, we and others have shown that Beclin1 plays an important role in viral replication and the innate immune signaling pathways. We previously used the cationic polymer polyethyleneimine (PEI) conjugated to mannose (Man) as a non-viral tool for the delivery of a small interfering (si) Beclin1-PEI-Man nanoplex, which specifically targets mannose receptor-expressing glia (microglia and astrocytes) in the brain when administered intranasally to conventional mice. To expand our previous reports, first we used C57BL/6J mice infected with EcoHIV and exposed them to combined antiretroviral therapy (cART). We show that EcoHIV enters the mouse brain, while intranasal delivery of the nanocomplex significantly reduces the secretion of HIV-induced inflammatory molecules and downregulates the expression of the transcription factor nuclear factor (NF)-kB. Since a spectrum of neurocognitive and motor problems can develop in people living with HIV (PLWH) despite suppressive antiretroviral therapy, we subsequently measured the role of Beclin1 in locomotor activities using EcoHIV-infected BECN1 knockout mice exposed to cART. Viral replication and cytokine secretion were reduced in the postmortem brains recovered from EcoHIV-infected *Becn1^+/−^* mice when compared to EcoHIV-infected *Becn1^+/+^* mice, although the impairment in locomotor activities based on muscle strength were comparable. This further highlights the importance of Beclin1 in the regulation of HIV replication and in viral-induced cytokine secretion but not in HIV-induced locomotor impairments. Moreover, the cause of HIV-induced locomotor impairments remains speculative, as we show that this may not be entirely due to viral load and/or HIV-induced inflammatory cytokines.

## 1. Introduction

The central nervous system (CNS) constitutes a viral reservoir comprising microglia/macrophages and astrocytes brain cell types, capable of establishing and maintaining hidden Human Immunodeficiency Virus (HIV) DNA, despite a local control of HIV replication by antiretroviral therapies (ART). Viral DNA contributes to a chronic state of neuroinflammation that can lead to the development of serious comorbidities such as neurological disorders and cognitive decline. About fifty percent of PLWH that are on long-term combined ART develop asymptomatic or mild neurocognitive impairments (NCI) [1,2,3,4]. NCI is typically characterized in neuropsychological tests as a weakness or loss in motor activities and a loss in memory [5,6]. Currently, there are no therapies to prevent NCI, and the number of cases continues to increase. The occurrence of NCI has been associated with the viral burden, neuroinflammation, and neurotoxic effects of HIV proteins produced by infected cells in the brain [7]. Therefore, the failure of cART in preventing NCI suggests a critical unmet need for an adjunctive therapeutic that could be successful in suppressing the glial reservoir and in reducing viral burden, neuroinflammation, and viral protein expression. This may hopefully result in a decreased occurrence of NCI.

We believe that Beclin1 is a promising and potential target therapeutic for suppressing the HIV-glial reservoir. The mammalian protein Beclin1, encoded by the *Becn1* gene, plays a central role in autophagy [8,9,10], an evolutionarily conserved transport pathway involved in the delivery of cytoplasmic material to the lysosomes for degradation [11]. We previously showed an increased expression of several autophagy-related proteins in the postmortem brains of HIV patients [12], while silencing the *Becn1* gene with small interfering (si) RNA significantly reduced HIV replication and the secretion of inflammatory molecules in viral-infected human astrocytes and microglia [13,14,15]. This action was mediated through the downregulation of the transcription factor nuclear factor (NF)-kB, as well as the downregulation of the ERK/JNK pathway in astrocytes and the STAT1/RS6 pathway in microglia [14]. Using mixed glia cultures recovered from autophagy-impaired *Becn1*^+/−^ deficient mice, we further showed a link between Beclin1 and the pathology of HIV in glia [9]. Using a cationic polymer, polyethyleneimine (PEI), conjugated to mannose (Man) as a tool for the delivery of a siBeclin1-PEI-Man nanoplex, we recently showed both brain delivery and target specificity of the mannosylated nanoplex to mannose receptor-expressing glia (microglia and astrocytes) after intranasal delivery in mice [16].

In this report, we used a conventional and an autophagy-related beclin1 (*Becn1*) knockout mouse model infected with EcoHIV to further decipher the role of Beclin1 in the underlying mechanism mediating HIV-associated neuropathology. The chimeric retrovirus EcoHIV was developed by Dr. David Volsky and his laboratory [17]. EcoHIV uses the murine leukemia virus gp80 in place of the gp120 from HIV for cell entry through the catamino acid transporter, thereby switching the viral tropism from human to rodent [18]. Infection with EcoHIV in conventional mice was recently shown by us [19]. As shown by others, EcoHIV replicates primarily in CD4+ T lymphocytes, macrophages, and microglia. In the brain, the virus establishes a lifelong chronic infection that can lead to NCI and depression-like defects in infected animals, resembling chronic suppressed HIV infection in PLWH [20,21,22]. EcoHIV persistence in conventional mice causes low levels of latent/inducible provirus in CD4+ T lymphocytes and expressed virus in macrophages. The intracranial injection of EcoHIV into mice showed that the virus infects primarily macrophage/microglial cells and not neuronal cells in the brain [23,24]. EcoHIV-infected brain tissues express elevated levels of inflammatory cytokines, chemokines, and a spectrum of host antiviral factors [17,25], making this virus ideal for the current studies. Overall, the findings show that a transient inhibition of Beclin1 reduces viral-associated inflammatory responses in the brains of EcoHIV-infected conventional mice. Moreover, EcoHIV-infected *Becn1^+/−^* mutant mice confirmed the role of Beclin1 in the regulation of viral load and cytokine secretion but not in EcoHIV-induced locomotor impairments.

## 2. Materials and Methods

### 2.1. Characterization of siBeclin1-PEI Nanoplex

The siBeclin1-PEI nanoplexes were characterized by size, charge, and polydispersity using a nanoparticle tracking analysis and the Nanoparticle Tracking Video Microscope PMX-120 from Zeta View, Particle Metrix (Inning am Ammersee, Germany).

### 2.2. Animals

C57BL/6J (*Becn1^+/+^*) mice (stock # 000664) and the *Becn1^+/−^* mice (stock # 018429) were procured from The Jackson Laboratory (Bar Harbor, ME, USA) and bred in the animal facility at Florida International University. All animal experiments were carried out in accordance with the guidelines of the National Institutes of Health Guide for the Care and Use of Laboratory Animals and the Florida International University Institutional Animal Care and Use Committee under protocols (IACUC #20-007 and #18-007). The B6.129 × 1-Becn1tm1Blev/J mice were heterozygous for the *Becn1* gene and experienced impaired induction of autophagy [26,27]. Mouse genotyping of tail genomic DNA was performed to detect wild-type and *Becn1* knockout alleles via PCR amplification, as previously described by us [9].

### 2.3. Viral Infection

EcoHIV plasmid was provided by Dr. David Volsky [17]. Virus stocks were generated with the transfection of EcoHIV plasmid DNA into HEK293T cells, as previously described by the Volsky lab [17]. Male mice (*n* = 5–8/treatment) between the ages of 4 and 6 months were administrated 1 mL of either saline or 1  ×  10^6^ pg p24 units of EcoHIV via the intraperitoneal cavity with and without antiretrovirals. At the indicated timepoints described in the text, blood was collected from the submandibular vein and transferred in a 0.5 mL heparin tube. Serum was subsequently obtained through centrifugation. A combination of Lopinavir, Abacavir, and Atazanavir at 100 mg/kg were administered intraperitoneally every other day. The combination of antiretrovirals was selected based on our previous study [16].

### 2.4. Intranasal Administration of siBeclin1-PEI-Man Nanoplex into EcoHIV-Infected C57BL/6 Mice

Beclin1 siRNA (siBeclin1) (Cat#: sc-29797, Santa Cruz Biotechnology, Santa Cruz, CA, USA) consists of pools of three to five target-specific 19–25 nucleotide sequences. Beclin1 siRNA was administered intranasally in non-infected and EcoHIV-infected adult C57BL/6 mice using the mannosylated PEI (PEI-Man) (Cat#: 203-10G, Polyplus-transfection, New York, NY, USA) according to the manufacturer’s protocol and as previously described by us [16]). A subset of animals received 20 μL of PBS, while another group received 20 μL of 20 μg/kg (N/P = 8) of siBeclin1-PEI-Man via intranasal route. After indicated time points, mice were sacrificed, and the tissues recovered were frozen in liquid nitrogen. Half of the tissues were used for histology analysis and the other half were minced and used for biochemical analysis.

### 2.5. ELISA

Secretion of tumor necrosis factor alpha (TNF-α), interleukin (IL)-6, and monocyte chemotactic protein-1 (MCP-1) and regulated on activation normal T cell expressed and secreted (RANTES) were measured by ELISA (R&D Systems, Minneapolis, MN, USA). Viral load was quantified by p24 levels using HIV-Gag p24 protein ELISA (ZeptoMetrix, Buffalo, NY, USA). The optical density was read at A450 on a Synergy HTX plate reader (BioTek, Winooski, VT, USA).

### 2.6. Growth Factor Antibody Array

Expression profiles of 10 growth factors were measured semi-quantitatively in postmortem brain tissues using the Mouse Growth Factor Array C1 (catalog #: AAM-GF-1-8, Ray Biotech, GA, USA), Briefly, antibody array membranes were incubated with lysates overnight at 4 °C, followed by biotinylation and streptavidin labeling. Chemiluminescence signals were detected using ChemiDoc imaging system (Bio-Rad, Hercules, CA, USA).

### 2.7. Hematoxylin and Eosin (H&E) Staining

Postmortem tissues frozen in optimal cutting temperature (OCT) compound were sectioned at 12-micron thickness and stained with H&E. Briefly, tissues were exposed to xylene and rehydrated with absolute ethanol, 95 percent, and 70 percent ethanol, followed by staining with hematoxylin dye for 15 min prior to washing with distilled water. Subsequently, tissue sections were stained with eosin for 20 s, dehydrated with gradient ethanol after washing with tap water, and tissues were then cleared by xylene and mounted using mounting media for visualization. Images were acquired using a Zeiss inverted fluorescence microscope with a 560 Axiovision camera (Zeiss, Oberkochen, Germany). Representative images are shown at 20×.

### 2.8. Nissl Staining

Postmortem brain tissues frozen in OCT were sectioned at 12-micron thickness and stained with cresyl violet acetate solution. Briefly, sections were rewarmed at room temperature for 30 min, then exposed to xylene and immersed in 100%, 95%, and 75% ethanol and distilled water, and then stained with a cresyl violet solution for 20 min. Sections were washed in distilled water then immersed in 75%, 95%, and 100% ethanol. Tissues were then cleared by xylene and mounted using mounting media for visualization. Images were acquired using an inverted fluorescence microscope with a 560 Axiovision camera (Zeiss, Germany). Representative images shown at 20× and 40×.

### 2.9. Behavioral Tests

Locomotor behavioral assessments were measured in animals at week 1 post-infection using the grip strength (gs) test, the accelerating rotarod (rr) test, and the horizontal bar (hb) test, as previously described by us [16]. After 10–15 min of acclimation, each animal was placed on a rotating rod with either constant rotation or steady acceleration. The latency to fall in seconds was recorded. Following the rr test, each animal’s paws were placed on the wire grid of the gs test. The animal naturally holds onto the wire while the tail is gently pulled backward. The maximum strength of the grip prior to grip release was recorded. After the gs test, each animal was placed on the hb test, consisting of a 2 mm and a 4 mm bar, to measure the strength of the forelimbs [28,29,30].

### 2.10. Statistical Analysis

Results are reported as mean  ±  SEM of 3–5 independent experiments. Data were analyzed using analysis of variance (ANOVA), followed by the post hoc Tukey’s test for multiple comparisons (GraphPad Software, Inc., La Jolla, CA, USA). An alpha level (*p*-value) of <0.05 was considered significant.

## 3. Results

### 3.1. Transient Inhibition of Beclin1 Reduces Viral Production and Attenuates Secreted Viral-Induced Inflammatory Molecules in EcoHIV-Infected Mice Co-Administered with Antiretroviral Drugs

We characterized siBeclin1 nanoparticles by size, charge, and polydispersity using the Nanoparticle Tracking Video Microscope PMX-120, which captures the Brownian motion of each particle in the video (Figure 1A). Based on the different diffusion movements of large and small particles in the surrounding liquid, the hydrodynamic diameter of the particles was determined. We evaluated all components of the formulation, including the mannosylated PEI alone and the siBeclin1-PEI-Man nanoplex. The actual size of the PEI-Man alone was around 175 nm, and the size of the siBeclin1 nanoplex was around 165 nm (Figure 1A). The evaluation of all components in the nanocomplex showed that the formation of siBeclin1-PEI-Man slightly decreased the average size of the particles, suggesting more a compact structure, presumably due to the decreasing overall positive charge (11.83 versus 2.32) of the PEI molecules that interacted with the negatively charges siRNA molecules. The size and charge distribution of PEI-Man alone is shown in Figure 1B(a,c), and the size and charge distribution of siBeclin1-PEI-Man nanoplex is shown in Figure 1B(b,d).

Male C57BL/6J mice (*n* = 5–8/treatment) were infected with a single dose of EcoHIV or an equal volume of PBS, as described in the methods and illustrated in Figure 1C. After three weeks of infection, the animals received a daily intraperitoneal injection of combined ART, while a subset of ART-treated animals received an intranasal administration of siBeclin1-PEI-Man nanoplex every other day. After 5 days, the animals were sacrificed, and the silencing efficiency of siBeclin1 was confirmed via Western blotting and showed a significant downregulation of the Beclin1 protein in the postmortem brains of the mice exposed to siBeclin1-PEI-Man (Figure 1D). The autophagy-related genes were also measured using RNA isolated from the corresponding postmortem brain tissues, and a decreased expression of *Becn1* along with several other genes in the autophagy pathway was confirmed through RT-PCR (Appendix A). The brains recovered from mice infected with EcoHIV showed a significant viral load that was subsequently decreased by about 1.3-fold after exposure to combined ART, while co-exposure to siBeclin1-PEI-Man had minimal effect on viral replication when compared to combined ART alone (Figure 1E). Interestingly, the intranasal administration of the siBeclin1-PEI-Man nanoplex significantly decreased the production of IL-6 (Figure 1F), TNF-α (Figure 1G), and RANTES (Figure 1I) when compared to brains exposed to combined ART alone. Brain homogenates were also used to semi-quantitatively detect the secretion of 10 mouse growth factor proteins (Figure 1J). The levels of granulocyte-macrophage colony-stimulating factor (GM-CSF), macrophage colony-stimulating factor (M-CSF), epidermal growth factor (EGF), and insulin-like growth factor-I (IGF-I) were significantly decreased in the brain tissues co-exposed with the siBeclin1-PEI-Man nanoplex as compared to those exposed to combined ART alone by 2.1-, 3.0-, 3.8-, and 2.0-fold, respectively.

To elucidate a potential mechanism responsible for the reduction of cytokines and growth factor responses by the siBeclin1-PEI-Man nanoplex in postmortem brain tissues, we measured the protein expression of the nuclear factor kappa B (NF-κB) and showed a significant decrease in expression level (Figure 1K). Additional studies using a transcription factor array confirmed the downregulation of NF-kB at the mRNA level as well as the downregulation of the transcription factor AP-2 (Appendix A). In a separate experiment using glia cell cultures recovered from pups, we further observed reduced expression of NF-κB in the cytoplasm of glia derived from *Becn1^+/^*^−^ when compared to control glia in the presence or absence of HIV Tat (Appendix A). The overall findings confirm that the protein Beclin1 not only regulates the secretion of pro-inflammatory cytokines but also growth factors induced by EcoHIV (GM-CSF, EGF) and EcoHIV + ART (M-CSF, IGF).

### 3.2. Minimal Anti-Inflammatory Action in Peripheral Organs after Transient Inhibition of Beclin1

We previously showed the detection of the nanoplex in murine lungs after intranasal delivery [16]. On that note, we determined whether the transient inhibition of Beclin1 could deliver an anti-inflammatory action in the lungs and additional organs, including the liver and kidney. The secretion of MCP-1 and RANTES were significantly increased by eight-and five-fold, respectively, in the lungs recovered from EcoHIV-infected mice (Figure 2A). Exposure to ART alone, irrespective of the siBeclin1-PEI-Man nanoplex, caused a decrease in secretion in MCP-1 but not in RANTES. (Figure 2A). Moreover, the secretion of IL-6 was decreased by 1.5-fold in the lungs recovered from mice co-exposed with the siBeclin1-PEI-Man nanoplex as compared to combined ART alone (Figure 2A). The secretion of cytokines in the liver and kidney was not significantly induced after EcoHIV infection (Figure 2B,C), despite the detection of the p24 Gag protein in these tissues (Appendix A). However, the secretion of MCP-1, TNF-α, and RANTES were increased in the liver, while the secretion of IL-6 was increased in the kidneys recovered from mice co-exposed with ART and the siBeclin1-PEI-Man nanoplex when compared to tissues exposed to ART alone (Figure 2B,C).

We also investigated potential morphological changes in these organs using a histological analysis (Figure 2D). The H&E staining showed no detectable tissue damage in the lung, liver, and kidney after the intranasal delivery of the siBeclin1-PEI-Man nanoplex (Figure 2D). The postmortem lung tissues recovered from mice exposed to the siBeclin1-PEI-Man nanoplex showed improved integrity when compared to EcoHIV-infected lungs with and without ART (Figure 2D). Overall, a slight increase in MCP-1, RANTES, and TNF- α was measured in the liver, and a small increase in TNF- α was measured in the kidney after siBeclin1 nanoplex delivery. Although speculative, this increase in cytokines may be due to ART-associated toxicity as a result of a decrease in autophagy. The antiretrovirals used in this study are principally metabolized and eliminated by the liver, where the autophagy pathway plays a key role in their clearance. It has been reported that autophagy is increased by several protease inhibitors [31]. However, the pharmacological inhibition of autophagy can exacerbate the antiretroviral-associated hepatotoxicity [32]. However, more studies are needed to confirm our speculation and to determine if the siBeclin1 nanoplex targets the liver directly or indirectly via cellular or extracellular processes. Future studies are needed to discern the consequences of increased cytokines in the liver and kidney with the transient inhibition of Beclin1 in the context of HIV.

### 3.3. EcoHIV-Infected Becn1^+/−^ Transgenic Mice Confirmed the Role of Beclin1 in the Regulation of HIV-Induced Cytokine Secretion but Not in HIV-Induced Locomotor Impairments

Since a spectrum of neurocognitive and motor problems can show up in PLWH despite suppressive antiretroviral therapy, we measured the role of Beclin1 in HIV-associated locomotor impairments using EcoHIV-infected BECN1 knockout mice exposed to combined ART. To further our study on the function of Beclin1, we used a mouse model with a monoallelic deletion mutant of Beclin1. *Becn1^+/−^* are backcrossed in C57BL/6J (*Becn1^+/+^)* mice, and in this study, C57BL/6J were used as a control (Figure 3A). The survival rate was 100%, irrespective of mouse strain (Figure 3B). The quantification of the EcoHIV RNA genome copies in the serum recovered from the mice infected with EcoHIV showed minimal signs of infection after weeks 1 and 2 post-infections (<200 RNA copies/mL), and at 3-weeks post-infection, about 596 RNA copies/mL were detected in the serum recovered from the *Becn1^+/+^* mice and 532 RNA copies/mL in the serum recovered from the *Becn1^+/−^* mice (Figure 3C). After 4 weeks post-infection, approximately 978 RNA copies/mL were detected in the serum recovered from the C57BL/6J control mice and about 1014 RNA copies/mL in the serum recovered from the *Becn1^+/−^* mice (Figure 3C). At the indicated time points, behavioral assessments were measured using the grip strength (gs) test, the horizontal bar (hb) test, and the rotarod (rr) test (Figure 3D). The gs test measures motor weakness in the forelimb and hind limb, the hb test measures strength, particularly of the forelimbs, and the rr test measures motor coordination and balance [33,34,35]. At seven days post-infection, the mice exhibited a decrease in the scores on the hb test when compared to the saline-treated mice, irrespective of strain (Figure 3D). The performance on the gs test remained relatively unchanged, except for a significant drop on day 14 post-infection. The performance on the rr test showed a significant decrease on day 28 for the *Becn1*^+/−^ strain but no statistically significant decrease was measured for the control mice (Figure 3D). On day 28 post-infection, the EcoHIV-infected mice showed extreme difficulty in grasping the bars, irrespective of strain (Figure 3D).

After the behavioral assessments, the mice were sacrificed, and the brains (hippocampus) recovered postmortem were used for Nissl staining to analyze neuronal morphology (Figure 3E). The histological staining showed the presence of pyramidal- and non-pyramidal-shaped neurons in the brain tissues recovered from C57BL/6J control mice treated with saline, and the distribution and morphology of the neurons were well maintained. The brain tissues recovered from C57BL/6J control mice infected with EcoHIV for 7 days showed a reduced number of Nissl bodies with less numbers of neurons, while C57BL/6J control mice infected with EcoHIV for 28 days showed a slight increase in glial cells, although the morphology of the neurons was well maintained. Likewise, the tissues recovered from *Becn1*^+/−^ mice treated with saline showed a well-maintained distribution and the presence of pyramidal- and non-pyramidal-shaped neurons, with minimal loss of cell bodies. No apparent differences were observed between the different strains when treated with saline. On the other hand, the brains recovered from *Becn1*^+/−^ mice infected with EcoHIV for 7 days showed the presence of more astrocytes, which are commonly round-shaped and in pairs (A), and microglia, which are usually round, elongated, or comma-shaped (M). Similarly to day 7, the tissue recovered from *Becn1*^+/−^ mice infected with EcoHIV for 28 days showed increased numbers of astrocytes (A) and microglia (M) when compared with *Becn1*^+/−^ mice treated with saline. Neuron swelling and vacuolation (long arrow) were also evident. The presence of glial cells appeared higher in *Becn1*^+/−^ mice infected with EcoHIV when compared to tissues from similarly infected C57BL/6J control mice. However, no neuronal shrinkage or loss was detected, irrespective of strain or infection. Taken together, these data suggest that EcoHIV infects both C57BL/6J conventional and autophagy-reduced mouse models. Furthermore, the infected mice exhibited similar locomotor impairments, irrespective of Beclin1.

### 3.4. Effect of Combined ART on Viral Load in Brain Recovered from Becn1^+/−^ versus C57BL/6J (Becn1^+/+^) Mice

Lastly, we evaluated the response of combined ART on viral load in *Becn1^+/−^* and C57BL/6J control mice after infection with EcoHIV (illustrated in Figure 4A). After one week of infection, a cocktail of ART consisting of emtricitabine, ritonavir, and atazanavir at 100 mg/kg was administered every other day for 21 days (Figure 4A). After 3 weeks of ART exposure, the mice were sacrificed, and the brains recovered at the necropsy were homogenized. The viral titer showed a 2.6-fold higher level of p24 in the EcoHIV-infected C57BL/6J control mice when compared to the viral-infected *Becn1^+/−^* mice. Moreover, exposure to ART caused a greater response in the C57BL/6J control animals (Figure 4B: black dots) and drastically reduced viral load by 2-fold when compared to *Becn1^+/−^* mice, which caused no significant reduction in viral load (Figure 4B: brown dots). Increased secretion of inflammatory cytokines, IL-6, TNF-α, and MCP-1, was measured in EcoHIV animals irrespective of the murine strain (Figure 4C), although the secretion of IL-6 was about 1.4-fold higher in the brains recovered from the C57BL/6J control mice (Figure 4C: black dots) when compared to the EcoHIV-infected *Becn1^+/−^* mice (Figure 4C: brown dots). This was also explored in vitro using glia cell cultures derived from the *Becn1^+/−^* and C57BL/6J control mice (Appendix A). The other half of the brain hemisphere was used for histology (Figure 4D). Nissl staining was performed to assess neuronal loss, and the labeled brains recovered from *Becn1*^+/−^ and C57BL/6J control mice showed a well-maintained distribution and morphology of the neurons, irrespective of the strain. However, the brains recovered from *Becn1*^+/−^ mice infected with EcoHIV showed the presence of more glia and a reduced number of Nissl bodies with fewer neurons when compared to C57BL/6J control mice infected with EcoHIV. The ART treatment did not show an apparent improvement in the number and morphology of neurons in both the *Becn1*^+/−^ and C57BL/6J control mice. Overall, the results show that the stable deletion of Beclin1 controls EcoHIV replication, irrespective of ART. Furthermore, the anti-inflammatory action of ART was effective in brains, irrespective of the murine strain.

## 4. Discussion

Our group and others have shown that the replication of HIV and the secretion of a set of inflammatory molecules in the brain are supported by select proteins of the autophagy pathway [9,15,36,37]. The mammalian protein Beclin1 is of importance as it is required for the initiation step of autophagy and is responsible for recruiting other proteins to the PI3K complex [38]. We showed that targeting the autophagy-Beclin1 protein (transiently with siRNA) in HIV-infected glial cell cultures reduces viral replication and viral-induced inflammatory molecules, while co-exposure to ART further reduces these actions [16]. This thus illustrates the importance of targeting Beclin1 as an adjunctive therapeutic approach against HIV in brain reservoirs. Additionally, we have reported the successful delivery of the nanoplex and the presence of siBeclin1 in the brain and lung tissues of uninfected mice after intranasal delivery using several techniques, including RT-PCR, immunofluorescence, and in vivo imaging system (IVIS) [14,16]. Here, we used an EcoHIV-infected mouse model administered intranasally with the siBeclin1-PEI-Man nanoplex to further confirm our findings.

Concurring with our previous in vitro studies, here we showed a reduction in viral load and in the secretion of the cytokines TNF-α, IL-6, and MCP-1. It is worth mentioning that the in vivo findings were not as robust as in the in vitro studies [16], as we were not able to detect a synergistic therapeutic effect after co-exposure with different ARTs and the nanoplex. The rationale for choosing different ART regimens was based on their relevance in the clinical setting and on a previous report by us showing a synergistic interaction between protease and/or integrase inhibitor-based antiretrovirals and siBeclin1 [16]. This thus highlights that what occurred in the control condition did not necessarily translate to the condition of a complex organism. Like cytokines, growth factors are important molecules for cell communication, and the dysregulation of growth factors can be implicated in many pathologies [39,40,41]. We reported that the growth factors GM-CSF, M-CSF, VEGF-A, and EGF in postmortem brains were upregulated by EcoHIV alone and after exposure with ART, while the intranasal administration of the nanoplex caused a significant decrease in the above-mentioned growth factors and IGF-1. Although most growth factors, including GM-CSF, have been shown to have a protective function in neurons [42], when it comes to HIV replication, the findings can be controversial. Studies have reported that GM-CSF could upregulate the LTR-driven transcription of HIV-1 through modulation of the transcription factors NF-κB and SP1 [43]. Reports by others have demonstrated that in vitro, GM-CSF can suppress HIV replication in human monocyte-derived macrophages [44], while others have shown that GM-CSF does not affect HIV replication in monocyte lineages [45]. However, when administrated to HIV-positive patients at any stage of the disease, without any antiretroviral therapy, they appeared to increase HIV activity [46]. IGF-1 is another growth factor known to exhibit neuroprotective functions [47]. At physiological concentrations, IGF-1 was shown to have inhibitory effects on HIV replication in cultured cord blood mononuclear cells and in chronically HIV-infected U937 cells, while EGF demonstrated no inhibition of HIV replication [48]. In terms of mechanism, the transcription factor NF-κB was significantly downregulated (at mRNA and protein levels) and the transcription factor AP-2 was significantly downregulated (at mRNA levels) after the administration of the nanoplex. This agrees with reports by others showing a specific binding site for AP-2 and NF-κB in the HIV-1 long terminal repeat which modulates the HIV enhancer function [49].

Overall, we showed that the intranasal administration of the siBeclin1-PEI-Man nanoplex offers many advantages over other common routes as it is non-invasive and easily accessible for the administration of drugs with the possibility to bypass the BBB, a limiting factor for targeting brain reservoirs and HIV-associated neurological disorders in the CNS [50,51,52]. Intranasal delivery systems can also result in systemic exposure to achieve a wide range of therapeutic effects. On that note, we also measured the effect of the nanoplex on the secretion of inflammatory molecules in other organs, including the lungs, liver, and the kidney. Others have shown traces of HIV, DNA, and RNA in the liver and the kidney recovered from EcoHIV-infected mice [18], while studies investigating the mechanisms of chronic obstructive pulmonary disease (COPD) reported EcoHIV in alveolar macrophages [53]. The latter showed a significant increase in IL-6 in smoke-exposed EcoHIV-infected mice [53], suggesting that maybe a greater factor alongside EcoHIV is required to compromise host immune responses in the lungs, kidney, and liver.

Here, we showed EcoHIV infection in the brains of conventional and autophagy-defective mice in vivo to further decipher the role of Beclin1 in the underlying mechanism mediating HIV-associated neuropathology. We used a mouse model with a stable deletion in the BECN1 gene, which enabled us to investigate the role of Beclin1 in HIV-associated locomotor impairments. The impaired muscle strength activity in EcoHIV-infected mice correlated with an increase in viral replication, irrespective of the strain. The rationale for using motor behavioral assessments is based on the neurocognitive impairment seen in HIV, which includes a progressive loss/weakness in limbs and motor skills [6,54]. In a previous report by us [30], we used similar mouse models exposed to the HIV Tat protein via intracranial injection for a short period of time. We reported an increase in the performance of the horizontal bar in Tat-exposed *Becn1^+/−^* when compared to Tat-exposed control animals [30]. Here, our findings showed a decrease in the performance of the horizontal bar and rotarod in mice infected with EcoHIV, irrespective of the strain. Although we did not observe changes in sensorimotor coordination, an increase in gliosis was detected in *Becn1^+/−^*-derived brains. Furthermore, brain histopathology showed greater gliosis, increased neuron swelling, and vacuolation in the brains recovered from EcoHIV-infected mice when compared to the saline-treated group. This is consistent with previous reports of memory impairment, along with hippocampal dysfunction and synaptodendritic injury, reported in wild-type mice infected with EcoHIV [55]. The EcoHIV-infected brains recovered from *Becn1^+/^*^−^ mice further highlighted that Beclin1 is cytoprotective in neurons, as lower viral replication and neuroinflammation did not result in less neuronal damage when compared to the EcoHIV-infected C57BL/6J control *Becn1^+/+^* mice. Interestingly, the brains recovered from C57BL/6J control mice exhibited higher levels of p24 Gag than *Becn1^+/−^* mice, suggesting that the glial cells may be using the autophagy pathway as a mechanism to clear pathogens. The reduced infectivity of EcoHIV in *Becn1^+/−^* mice indicates that the infection efficiency is different between the two strains. These differences could be explained by the levels of NF-kB noticed in the cytoplasm versus the nucleus after exposure to the HIV protein Tat. We have reported earlier that reducing autophagy with siBeclin1 attenuates the secretion of pro-inflammatory molecules and viral replication through the inhibition of the transcription factor NF-κB and Ca^2+^ signaling pathways in vitro [12,15,16]. Likewise, studies using mixed glia culture derived from *Becn1^+/−^* mice showed a decrease in the secretion of inflammatory molecules after treatment with the HIV protein Tat when compared to similarly treated glia derived from C57BL/6J control mice [9]. Exposure with combined ART showed a reduction in brain viral titer in C57BL/6J control mice when compared to the EcoHIV-only group, and this reduction was not seen in the *Becn1*^+/−^ mice. The *Becn1^+/−^* mice exposed to ART did not exhibit a robust decrease in inflammatory secretion when compared to the C57BL/6J control mice, which was further explored using glia cell cultures derived from the Becn1^+/−^ and C57BL/6J control mice. The overall findings are suggestive of the role of Beclin1 in mediating HIV-induced neuroinflammatory molecule secretion.

In conclusion, our findings showed the potential use of targeting the autophagy protein Beclin1 as an effective adjunctive therapy in combination with antiretrovirals for the attenuation of HIV infection and HIV-induced inflammatory molecules in the brain. Concurring with others, our results indicate that Beclin1 is cytoprotective in neurons. However, here we have shown for the first time that *Becn1*^+/−^ mice can be infected with EcoHIV and that the protein Beclin1 is associated with the replication and neuroinflammation induced by EcoHIV, making this protein a viable target to prevent pathophysiology associated with HIV in infected patients. However, in terms of therapeutics, the silencing of *Becn1* should be targeted against brain reservoirs and not neurons, as Beclin1 is important for the induction of autophagy and for neuronal survival. Limitations to the study include the use of male mice, while female mice were used for breeding. We do recognize the importance of using both sexes in studies related to HIV, and ongoing studies in the laboratory are using both female and male animals. Secondly, the lack of a mass spectrometry detection method to confirm the active transport of ART across the BBB was a limitation. Lastly, it should be noted that in vivo and in vitro cells have different environments, and some discrepancies are always noted when presenting findings.

## Figures and Tables

**Figure 1 viruses-15-01923-f001:**
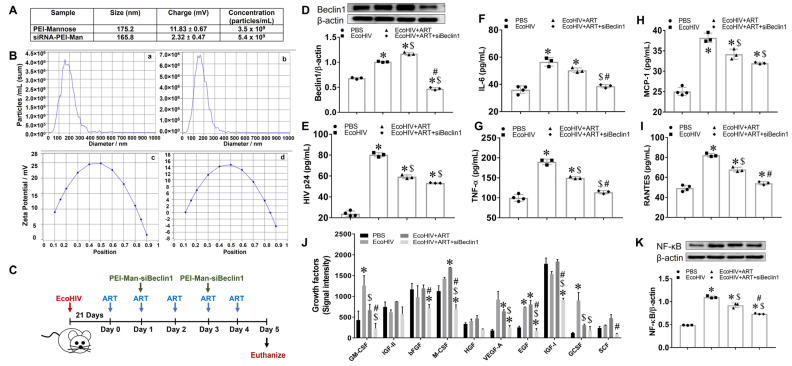
Therapeutic efficacy of siBeclin1-PEI-Man in EcoHIV-infected mice. (**A**) Characterization of nanoformulated siBeclin1. (**B**) Typical size distribution for PEI (**a**) and siRNA-PEI (**b**), and zeta potential distribution for PEI (**c**) and siRNA-PEI (**d**). (**C**) Schematic illustration of EcoHIV infection and intranasal administration of siBeclin1-PEI-Man. (**D**) Beclin1 protein expression was measured via Western blotting and normalized using the housekeeping protein β-actin. (**E**) Viral protein p24 and the cytokines (**F**) IL-6, (**G**) TNF-α, (**H**) MCP-1, and (**I**) RANTES were measured in postmortem brain tissues recovered from PBS- and EcoHIV-infected mice using ELISA. (**J**) The expression profiles of growth factors in postmortem brain tissues were measured using mouse growth factor antibody array. (**K**) NF-kB protein expression was measured via Western blotting and normalized using the housekeeping protein β-actin. (*p* < 0.05, *—vs. PBS, $—vs. EcoHIV, #—vs. EcoHIV + ART).

**Figure 2 viruses-15-01923-f002:**
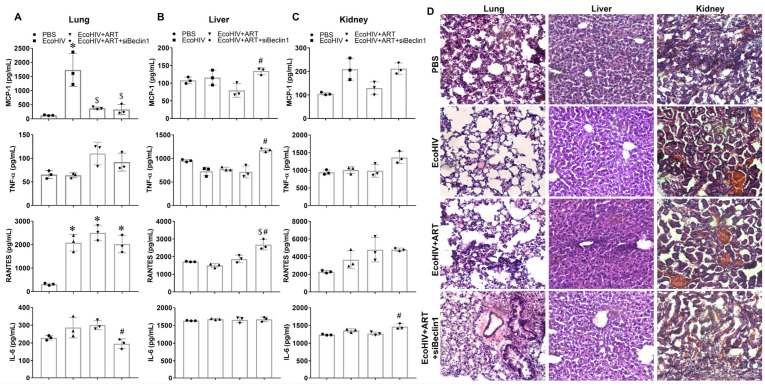
Secretion of inflammatory molecules and histological analysis in peripheral organs recovered at necropsy from PBS- and EcoHIV-infected mice. (**A**) Lung, (**B**) liver, and (**C**) kidney lysates were used to measure MCP-1, TNF-α, RANTES, and IL-6 by ELISA (*p* < 0.05, *—vs. PBS, $—vs. EcoHIV, #—vs. EcoHIV + ART). (**D**) Corresponding tissue sections of lung (**left panel**), liver (**middle panel**), and kidney (**right panel**) were used for histological analysis using H&E staining.

**Figure 3 viruses-15-01923-f003:**
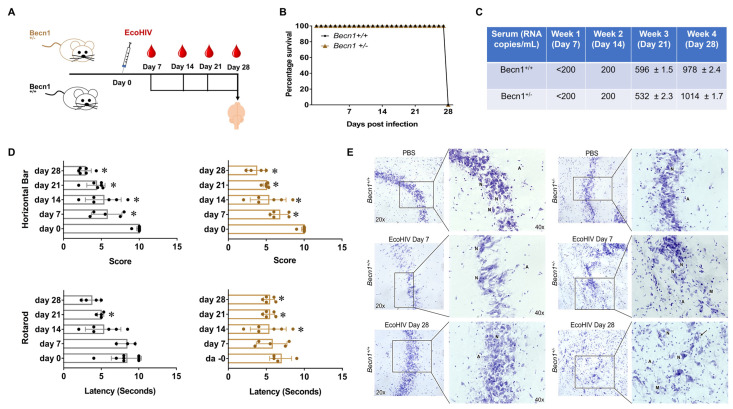
EcoHIV-infected C57BL/6J control (*Becn1^+/+^*) and *Becn1^+/−^* mice with minimal fatality rates. (**A**) Schematic illustration of EcoHIV infection and weekly blood collection from the submandibular vein. (**B**) Survival curve after infection with EcoHIV in C57BL/6J control (*Becn1^+/+^* = black) and *Becn1^+/−^* (=brown) mice. (**C**) Table indicates viral RNA copies per mL of p24 from serum collected at days 7,14, 21, and 28 post-EcoHIV infection. Numbers were obtained using LTR real-time PCR. (**D**) Horizontal bar behavior testing was performed on days 0, 7,14, 21, and 28 post-EcoHIV infection in C57BL/6J control *Becn1^+/+^* (left, black) and *Becn1^+/−^* (right, brown) mice. Each score represents the combined duration of a mouse on the 2 mm and 4 mm bar (*p* < 0.05, * vs. Day 0). Grip strength behavior testing was performed on days 0, 7,14, 21, and 28 post-EcoHIV infection in C57BL/6J control *Becn1^+/+^* (left, black) and *Becn1^+/−^* (right, brown) mice. Peak tension was normalized by the weight of each mouse (*p* < 0.05, * vs. Day 0). (**E**) Nissl staining of C57BL/6J control *Becn1^+/+^* (left) and *Becn1^+/−^* mice (right) brains, which were removed postmortem after exposure to saline (top panel) or EcoHIV for 7 days (middle panel) or 28 days (bottom panel). Images were acquired using an inverted fluorescence microscope with a 560 Axiovision camera at 20× and 40× magnifications (Zeiss, Germany). Neurons are indicated by N, astrocytes indicated by A, and microglia by M.

**Figure 4 viruses-15-01923-f004:**
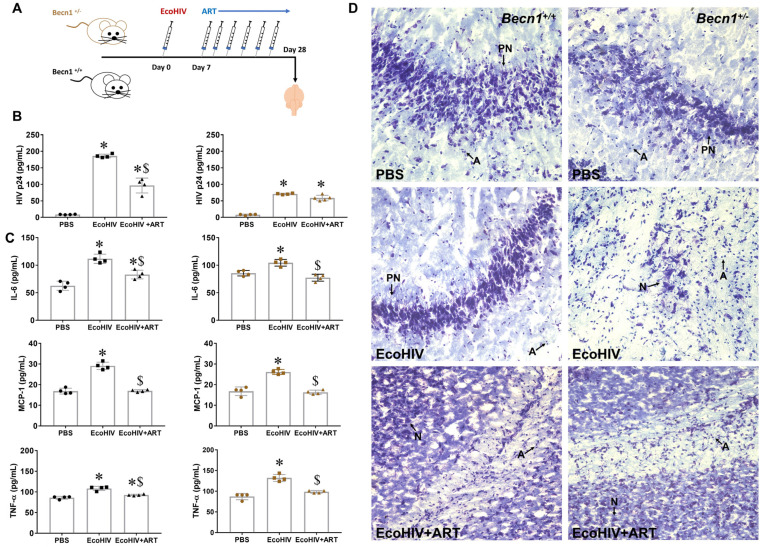
Treatment with antiretroviral therapy in *Becn1^+/−^* and C57BL/6J control (*Becn1^+/+^*) mice infected with EcoHIV. (**A**) Schematic illustration of EcoHIV infection and ART administration. (**B**) Viral protein, p24, and (**C**) the cytokines, TNF-α, IL-6, and MCP-1, were measured using ELISA in brains recovered at necropsy from C57BL/6J control *Becn1^+/+^* (left, black) and *Becn1^+/−^* (right, brown) mice (*p* < 0.05, *—vs. saline, $—vs. EcoHIV). (**D**) Nissl staining of C57BL/6J control *Becn1^+/+^* (left) and *Becn1^+/−^* mice (right) brains removed postmortem after exposure to saline (**top panel**), EcoHIV (**middle panel**), and EcoHIV and ART (**bottom panel**) for 28 days. Images were acquired using an inverted fluorescence microscope with a 560 Axiovision camera at 20× and 40× magnifications (Zeiss, Oberkochen, Germany). Purkinje neurons are indicated by PN; neurons are indicated by N; astrocytes indicated by A.

## Data Availability

The data presented in this study are available in the article or Appendix A.

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
