# Peer review of "Implication of the Autophagy-Related Protein Beclin1 in the Regulation of EcoHIV Replication and Inflammatory Responses"

_viruses, 2023, doi:10.3390/v15091923_

Round 1
Reviewer 1 Report
The manuscript by Rodriguez et al entitled, “Implication of the Autophagy-related protein Beclin1 in the regulation of EcoHIV replication and cytokine secretion, but not in HIV-induced locomotor impairments addresses potentially important roles for Beclin-1 in HIV-associated neurocognitive disorder in people with HIV. They utilize the frequently used EcoHIV mouse model in combination with cART to assess the impact of monoallelic KO of becln1.
The manuscript is extremely well-written. The introduction provides background and rationale for the current studies and supports the need for these investigations.
The data are very strong and support the claims made by the authors. These data add to our understanding of potential contributors to HAND. Mechanism(s) through which Beclin-1 regulates these processes in addition to NfkB, remain unclear.
How was blood taken form mice?
Please add rationale for the choice of the two different cART regimens.
Authors recognize the limitations of the study including only using male mice and lack of confirmation of cART reaching the brain. There is no mention of why female mice were not used.
Author Response
We would like to thank the reviewers for their insightful comments and have responded to each in blue font in the body of the manuscript. We were pleased with the overall outcome and enthusiasm among the reviewer panel and feel that the comments and suggested provided by the reviewers strengthened the manuscript.
REVIEWER 1:
Comment 1: How was blood taken from mice?
Response: The following sentence was added to the Materials and methods: “At indicated timepoints described in the text, blood was collected from the submandibular vein and transferred in a 0.5 mL heparin tube. Serum was subsequently obtained by centrifugation”.
Comment 2: Please add rationale for the choice of the two different cART regimens.
Response: The following paragraph was added to the Conclusion: “The rationale for choosing different ART regimen was based on their relevance in the clinical setting, and based on previous report by us, showing a synergistic interaction between protease and/or integrase inhibitor-based antiretrovirals and siBeclin1 [16].
Furthermore, during the revision of the manuscript we realized that the cART regimen used for the in vivo studies was incorrectly described in the materials and methods section of the main manuscript. The paragraph was consequently revised, and it reads as follows in the Materials and methods: “A combination of Lopinavir, Abacavir, Atazanavir at 100 mg/kg were administered intraperitoneally every other day. The combination of antiretrovirals was selected based on our previous study [16].”
Comment 3: Authors recognize the limitations of the study including only using male mice and lack of confirmation of cART reaching the brain. There is no mention of why female mice were not used.
Response: We appreciate the comment and added the following sentences to our initial statement. “Only male mice were used in this study, while female mice were used for breading. We do recognize the importance of using both sexes in studies related to HIV, and ongoing studies in the laboratory are using both female and male animals.”

Reviewer 2 Report
Journal Viruses (ISSN 1999-4915)
Manuscript ID viruses-2553197
Type Article
Title Implication of the Autophagy-related protein Beclin1 in the regulation of EcoHIV replication and cytokine secretion, but not in HIV-induced locomotor impairments
Authors Myosotys Rodriguez , Florida Owens , Marissa Perry , Nicole Stone , Yemmy Soler , Rianna Almohtadi , Yuling Zhao , Elena V. Batrakova , Nazira El-Hage *
The manuscript titled “Implication of the autophagy-related protein Beclin1 in the regulation of EcoHIV replication….” by Rodriguez M et al, shows that the suppression of beclin1 expression, achieved by intranasal injection of siBeclin1 PEI nanoparticles conjugated with MAN, could be an additional therapeutic tool effective in limiting HIV replication and reducing the production of inflammatory molecules in the brain together with ART.
The study was conducted on C57BL/6 mice infected with EcoHIV, exposed to cART and treated with the siBeclin1 PEI nanoparticles via intranasal delivery. The authors show the significant reduction of the HIV replication and of the secretion of HIV-induced inflammatory molecules in mice with silenced Becn1 (Becn1+/-), thus sustaining the importance of beclin1 in the HIV infection.
Although the work is very interesting, some major changes are necessary to make it clearer and therefore publishable. Indeed, there are numerous inaccuracies which often make it difficult to understand the results.
Title: needs to be clarified and reformulated.
How do the authors demonstrate the silencing of the beclin1 gene in the brains of mice receiving the nanoparticles? Whether the proof is in Figure 1D, it is unclear. The siBeclin1 control without EchoHIV is missing: it should be shown. The description needs to be improved, both in the text and in the caption of the figure.
In Becn1 +/- mice, have been observed effects on autophagy in the brain and other organs due to temporary silencing of beclin1?
Figure 1 A, B: Was the characterization of siBecn1 nanoparticles performed before inoculation or in the brain of mice after inoculation? Were the nanoparticles resuspended in the same medium in which they were administered into the nose during characterization? Which medium?
In the text, lines 210-220, the quotations from the panels in figure 1 are wrong or missing.
In Figure 2B and C, an increase of inflammatory factors is observed in the liver and kidneys of becn1+/- mice. How is it explained?
In Figure 4b and 4C the graphs should report the same numerical scale for an exact comparison.
In Figure 4B, smoothing the scale to 250pg/ml, a striking difference in p24 levels is observed between the becn1 +/+ and +/- mice. So the infection efficiency with EchoHIV seems different. Please explain.
In Figure 4C, standardizing the scale to 150pg/ml, instead there seems to be no difference between the becn1 +/+ and +/- mice.
In the Discussion: the text must be shortened and the figures must not be mentioned because it has already been done in the Results.
There is a paper (J Intern Med. 2017 May;281(5):422-432. doi: 10.1111/joim.12596. Epub 2017 Feb 16. Role of autophagy in HIV infection and pathogenesis. DOI: 10.1111/joim. 12596) showing that in HIV-infected Long term non-progressor patients, beclin1 is more highly expressed. In this work the control of HIV replication in the absence of ART was correlated with a more intense autophagic activity in PBMCs. Were the levels of other factors related to autophagy measured in the brains of mice?
Author Response
We would like to thank the reviewers for their insightful comments and have responded to each in blue font in the body of the manuscript. We were pleased with the overall outcome and enthusiasm among the reviewer panel and feel that the comments and suggested provided by the reviewers strengthened the manuscript.
REVIEWER 2:
Comment 1:Title: needs to be clarified and reformulated.
Response: We appreciate the suggestion; the title has been revised and now reads as follow “Implication of the Autophagy-related protein Beclin1 in the regulation of EcoHIV replication and inflammatory responses”.
Comment 2: How do the authors demonstrate the silencing of the beclin1 gene in the brains of mice receiving the nanoparticles? Whether the proof is in Figure 1D, it is unclear. The siBeclin1 control without EcoHIV is missing it should be shown. The description needs to be improved, both in the text and in the caption of the figure.
Response: Silencing of the beclin1 gene in the brains of mice receiving the nanoparticles was confirmed by RT-PCR and Western blotting. In addition to Figure 1D and our previous published work [2] showing downregulation of Beclin1 protein by Western blotting, we have data confirming the silencing of the Becn1 gene in EcoHIV-infected mice exposed to siBeclin1 nanoplex using RT-PCR assays. The results have been added to the supplemental figure 1.
Comment 3: In Becn1 +/- mice, have been observed effects on autophagy in the brain and other organs due to temporary silencing of beclin1?
Response: Our previous studies, using glia-derived from Becn1+/- mice, showed reduction in LC3 [3], while in vivo studies using Becn1+/- animals showed reduction in Beclin1 and LC3-II expression levels and increased p62/SQSTM1 levels in brain tissues [4]. In response to the comment, we have updated the Supplemental figure 1 to include findings that measured Autophagy-related genes using RNA extracted from brain tissues recovered postmortem from EcoHIV-infected mice, treated with cART, and/or exposed to siBeclin1 nanoplex in combination with cART. A statement has been added to the Results section and reads as follow: “Autophagy-related genes were also measured using RNA isolated from the corresponding postmortem brain tissues, and decreased expression of Becn1 along with several other genes in the autophagy pathway were confirmed by RT-PCR (Supplemental data 1).”In terms of other organs, ongoing studies in the laboratory are investigating effects on autophagy in the lungs, liver, kidney and spleen, due to temporary silencing of beclin1.
Comment 4: Figure 1 A, B: Was the characterization of siBecn1 nanoparticles performed before inoculation or in the brain of mice after inoculation? Were the nanoparticles resuspended in the same medium in which they were administered into the nose during characterization? Which medium?
Response: The characterization was performed before inoculation in the brain. The nanoparticles were resuspended in 10% glucose, the same medium in which they were administrated into the nose.
Comment 5: In the text, lines 210-220, the quotations from the panels in figure 1 are wrong or missing.
Response: We thank the reviewer for his/her careful observation. This was a clear oversight on our part, which we have corrected, accordingly.
Comment 6: In Figure 2B and C, an increase of inflammatory factors is observed in the liver and kidneys of becn1+/- mice. How is it explained?
Response: We have added the following paragraph to the text “A slight increase in MCP-1, RANTES and TNF was measured in the liver and a small increase in TNF was measured in the kidney, after siBeclin1 nanoplex delivery. Although speculative, the increase in cytokines may be due to ART-associated toxicity, as a result of a decrease in autophagy. The antiretrovirals used in this study are principally metabolized and eliminated by the liver, where the autophagy pathway plays a key role in clearance. It has been reported that autophagy is increased by several protease inhibitors [5]. While pharmacological inhibition of autophagy can exacerbate the antiretroviral-associated hepatotoxicity [6]. However, more studies are needed in order to confirm our speculation, and to determine if the siBeclin1 nanoplex target the liver directly or indirectly via cellular or extracellular processes”.
Comment 7: In Figure 4b and 4C the graphs should report the same numerical scale for an exact comparison.
Response: We have modified the scale bars corresponding to Figure 4B and Figure 4C to the same numerical scale.
Comment 8: In Figure 4B, smoothing the scale to 250pg/ml, a striking difference in p24 levels is observed between the becn1 +/+ and +/- mice. So, the infection efficiency with EcoHIV seems different. Please explain.
Response: We agree with the reviewer and have included the following paragraph in the conclusion “The reduce infectivity of EcoHIV in Becn1+/- mice indicates that the infection efficiency is different between the two strains. The differences could be explained by the levels of NF-kB noticed in the cytoplasm versus the nucleus after exposure to HIV protein Tat. We have reported earlier that reducing autophagy with siBeclin1 attenuated the secretion of pro-inflammatory molecules and of viral replication through the inhibition of the transcription factor, NF-κB, and Ca2+ signaling pathways in vitro[1,7,8]. Likewise, studies using mixed glia culture derived from C57BL/6J control Becn1+/- mice showed a decrease in the secretion of inflammatory molecules after treatment with the HIV protein Tat, when compared to similarly treated Becn1+/+ derived glia [3]”.
Comment 9: In Figure 4C, standardizing the scale to 150pg/ml, instead there seems to be no difference between the becn1 +/+ and +/- mice.
Response: While we do agree with the reviewer, we have shown a decrease of inflammatory cytokines with Becn1+/- mice [9]. However, the experiments in Figure 4C did not represent such difference. We have rephrased the sentences as follows:
“Furthermore, the anti-inflammatory action of ART was effective in brains, irrespective of the murine strain. Becn1+/- mice exposed to ART did not exhibit a robust decrease in inflammatory secretion when compared to C57BL/6J control Becn1+/+ mice, which was further explored using glia cell cultures-derived from Becn1+/- and control mice.”
Comment 10: In the Discussion, the text must be shortened, and the figures must not be mentioned because it has already been done in the Results.
Response: Per request of the reviewer, we have revised and shortened the discussion. Moreover, the figures are no longer mentioned.

Reviewer 3 Report
The manuscript by Rodriguez et al describes Beclin1 protein in the regulation of EcoHIV replication and cytokine secretion and possibility of a role in HIV induced locomotor impairments. They report that EcoHIV enters the mouse brain, while intranasal delivery of the nanocomplex significantly reduces the secretion of HIV-induced inflammatory molecules and downregulates the expression of the transcription factor nuclear factor (NF)-kB. Furthermore, using comparative studies in wild type versus BECN1 knockout mice infected with EcoHIV the results show viral replication and cytokine secretion were reduced in postmortem brain recovered from EcoHIV-infected Becn1+/− mice when compared to EcoHIV-infected Becn1+/+ mice. However, they found locomoter impairment to be comparable. The experiments are straight-forward, and the finding can serve as basis for a potential therapeutic application of Beclin1 pathway in HIV induced brain pathology. The manuscript can be enhanced by addressing the following points before acceptance.
1. Need to include experiments showing the delivery and presence of siRNA in the tissue.
2. A non-specific control siRNA should be included to demonstrate specificity.
3. Although the authors are basing their experiment to reflect the actual environment infection in the presence of drug therapy a control of EcoHIV infection with siRNA treatment only in the absence of drug therapy would be important.
4. Figures 1 and 2, the labels are too small for readers to read the manuscript smoothly.
5. Supplemental Figure, the mRNA chip analysis is labeled as Supplemental Figure 2. It should be Supplemental Figure 3.
Minor edition is required
Author Response
We would like to thank the reviewers for their insightful comments and have responded to each in blue font in the body of the manuscript. We were pleased with the overall outcome and enthusiasm among the reviewer panel and feel that the comments and suggested provided by the reviewers strengthened the manuscript.
REVIEWER 3:
Comment 1: Need to include experiments showing the delivery and presence of siRNA in the tissue.
Response: We have conducted extensive research on the delivery of siBeclin1 in the brain. We have consistently reported on the delivery of siRNA in the brain using several techniques. For example, live imaging analysis confirmed brain accumulation of siBeclin1 nanoparticles after 4, 24, 48, and 96 h post-administration [1]. We have quantitatively measured the presence of siBeclin1 (up to 120 h) in the olfactory bulb, frontal and midbrain, and the lungs after intranasal delivery, using stem-loop RT-PCR. 24 and 48 h timeframe coincided with the observed downregulation of Beclin1 protein expression levels in the postmortem brain tissues of mice intranasally administered with siBeclin1-PEI previously reported by us [7]. In summary, we have carefully optimized the concentration of the siBeclin1 nanoplex and confirmed the delivery efficiency in the brain. In this study, we focused on the effects of siBeclin1 in EcoHIV-infected animals. We have included a paragraph in the discussion that reads as follow:”Additionally, we have reported the successful delivery of the nanoplex and the presence of siBeclin1 in brain and lung tissues of uninfected mice, after intranasal delivery, using several techniques including RT-PCR, immunofluorescence, and In vivo Imaging System (IVIS)”.
Comment 2: A non-specific control siRNA should be included to demonstrate specificity.
Response: As with the comments regarding delivery and presence of siRNA in the tissue, we have previously shown delivery with control siRNA [1].
Comment 3: Although the authors are basing their experiment to reflect the actual environment infection in the presence of drug therapy a control of EcoHIV infection with siRNA treatment only in the absence of drug therapy would be important.
Response: While we agree with the reviewer, it was not feasible to add siRNA treatment only, as the infected animals would die. Secondly, siBeclin1 was meant to be used as an adjunctive therapy in combination with ART. Thus, for this complex study in EcoHIV-infected mice we focused on using siBeclin1 in the presence of ART exclusively, as it would be in a potential clinical setting.
Comment 4: Figures 1 and 2, the labels are too small for readers to read the manuscript smoothly.
Response: The size of the labels from Figure 1 and Figure 2 have been increased.
Comment 5: Supplemental Figure, the mRNA chip analysis is labeled as Supplemental Figure 2. It should be Supplemental Figure 3.
Response: We apologize for the oversight. Due to the addition of a new supplemental figure (Supplemental figure 1), the order of the figures has been revised.

Round 2
Reviewer 2 Report
In the present form the manuscript can be published.
Reviewer 3 Report
The authors response is adequate.